# The Stages and Grades of Periodontitis Are Risk Indicators for Peri-Implant Diseases—A Long-Term Retrospective Study

**DOI:** 10.3390/jpm12101723

**Published:** 2022-10-15

**Authors:** Mikiko Yamazaki, Kosaku Yamazaki, Yuh Baba, Hiroshi Ito, Bruno G. Loos, Keiso Takahashi

**Affiliations:** 1Department of Endodontics and Periodontics, Graduate School of Dentistry, Ohu University, 31-1 Misumido Tomita-machi, Koriyama 963-8611, Japan; 2Division of Periodontics, Department of Conservative Dentistry, School of Dentistry, Ohu University, 31-1 Misumido Tomita-machi, Koriyama 963-8611, Japan; 3Ohu University Dental Hospital Otorhinolaryngology, School of Dentistry, Ohu University, 31-1 Misumido Tomita-machi, Koriyama 963-8611, Japan; 4Division of Oral Pathology, Department of Oral Medical Science, School of Dentistry, Ohu University, 31-1 Misumido Tomita-machi, Koriyama 963-8611, Japan; 5Department of Periodontology, Academic Centre for Dentistry Amsterdam, University of Amsterdam and Vrije Universiteit, 1081 HV Amsterdam, The Netherlands

**Keywords:** periodontitis, classification of periodontitis, peri-implant disease, implant failure, peri-implantitis, lack of osseointegration, prevalence, risk indicators

## Abstract

The aim of this study is to evaluate the factors of implant failure in patients with periodontitis and their impact on the prognosis of having a peri-implant disease and/or implant failure. Data regarding 325 implants among 84 patients with periodontitis were retrospectively examined. Patients were classified by Stage (I, II, III, IV) and Grade (A, B, C), implant failures for peri-implant disease and lack of osseointegration. Clinical data, including implant- and patient-related variables were evaluated by principal components analysis (PCA) and two-step cluster analysis (CA). Survival and success rates were 96.3% and 87.1%, respectively. Prevalence of peri-implant disease was significantly higher in Stage IV patients (*p* < 0.05), and incidence of lost implant due to peri-implantitis was significantly higher in patients with bone augmentation (BA) (*p* < 0.05). PCA and CA revealed five of eleven variables and four clusters at patient level, and six of fourteen variables and three clusters at implant level. Stage and Grade are useful indicators for the development of peri-implant diseases in which BA and the number of implants are involved.

## 1. Introduction

Although the pathogenesis of peri-implant disease in patients with periodontitis remains unclear, rehabilitation with implant-supported fixed prostheses in comprehensive periodontal therapy is available. Oral implant treatment is not a panacea and implant failures occur due to many factors, including peri-implantitis, lack of osseointegration (LoO) and others [1,2]. The clinical definition of peri-implantitis has differed among conferences and workshops, suggesting an evidence-based definition of peri-implantitis remains controversial [3].

The prevalence of peri-implantitis, including early and late failures of implant treatment, has been reported [4]. Prevalence of peri-implantitis has been reported as 45% in a large-scale epidemiological study [5]. Canullo et al. have reported that risk of peri-implantitis was classified into three categories: surgical, prosthetic and bacteriological, with surgical skills of dentists being the most significant factor [6]. These results support the possibility that various risk factors regarding patient and/or implant may be involved in peri-implantitis [7,8]. The present study hypothesizes that the reliability of risk assessment of peri-implant disease may be improved by including participants with as few confounding variables as practical.

Peri-implantitis is a multifactorial disease [5,9]. Heterogeneity among study participants, quality of clinical records and patient-related factors, including history and severity of periodontitis, presence or absence of keratinized mucosa and smoking habit, are confounding variables that make it difficult to draw robust conclusions regarding peri-implant disease [10,11].

Based on the 2017 World Workshop on the Classification of Periodontal and Peri-implant Diseases and Conditions [3,12], periodontitis was sub-classified by Stage and Grade. To our knowledge, no study has evaluated the relationship between severity (Stage) and rate of progression (Grade) of periodontitis and peri-implant diseases based on the latest classification.

The aim of this study is to evaluate the factors of implant failure in patients with periodontitis and their impact on the prognosis of having a peri-implant disease and/or implant failure.

## 2. Materials and Methods

### 2.1. Study Design

The research protocol for this retrospective study was approved in 2018 by Ethics Board, Ohu University (reference number 222). Exclusion criteria were: (i) incomplete data—lack of radiograph taken after occlusal function; (ii) early failure—implant lost before occlusal function; (iii) non-compliance—behavior demonstrating uncooperative compliance to treatment; (iv) HbA1c ≥ 6.5%—uncontrolled diabetes mellitus. The periodontal conditions of patients were assessed and recorded by periodontists at initial examination, and diagnosed according to the latest classification of periodontal disease based on Stage (I, II, III and IV) and Grade (A, B, C) [3]. Inclusion criteria were: (i) patients were deemed compliant and enrolled as study participants only if they had kept good oral hygiene (O’ Leary’s Plaque Control Record ≤ 20%, routine follow-up visits); (ii) patients accorded with our periodontal therapy.

### 2.2. Periodontal and Implant Therapy

All patients had been comprehensively treated by experienced periodontists at Ohu University Dental Hospital. After their periodontal condition had been successfully treated non-surgically or surgically, probing pocket depths were reduced to less than 4 mm and plaque scores were <20%, the patients were enrolled for implant treatment and all gave written informed consent. All patients had received their implants between 2006 and 2018 and almost all by a single experienced periodontist (KT) who had at least 20 years’ experience of implant treatment for patients with periodontitis. All reported measurements were made during 2018–2020 by a trained postdoctoral student (MY). Data analysis ensured patient anonymity.

In all cases, treatment plans for implant therapy were designed using dental CT scan images. Two implant systems, POI (Physio Odentram Implants, POI EX^®^; Kyocera, Kyoto, Japan) and Bmk (Brånemark System^®^ Bmk; Nobel Biocare, Göteborg, Sweden), were used as described (Table 1). All implants were bone-level implants placed using open flap surgery. A two-stage surgical approach was performed in almost all cases, with an abutment placed in the second stage. Implants were placed according to manufacturer guidelines. In the second surgery, apically repositioning flap surgery with partial-thickness flap or free gingival graft procedure were additionally performed if there was lack of keratinized mucosa (<2 mm width) around the inserted implants. At implant sites with an atrophic ridge that required bone augmentation (BA), guided bone regeneration was performed before or at the time of implant placement using autogenous bone, hydroxyapatite and/or β-TCP using a Ti membrane (Jeil Medical, Seoul, Korea) or collagen membrane (Zimmer Biomet, Warsaw, IN, USA). At the atrophic maxilla site, either an osteotome indirect sinus lift without added bone graft or a sinus lift with a lateral window approach was performed. The healing period prior to restoration and loading of the implants was from 3 to 6 months after placement. The prosthetic reconstructions (all screw-retained) were performed by various periodontists.

### 2.3. Baseline and Follow-Up Measurements

Periodontal examination was performed at first visit (baseline). Intra-oral or panoramic radiographs were obtained at the time of implant placement and after placement of the restoration. Follow-up examinations were conducted by experienced periodontists. Clinical measurements were taken using a plastic periodontal probe (Nihon ShikenKougyou Co., Tokyo, Japan) set to measure under low pressure around implants and record to the nearest millimeter. A full periodontal chart at six points per tooth and implant was completed for each patient, recording probing pocket depths, bleeding on probing (BOP) and suppuration.

### 2.4. Radiographic Parameters

Intra- or extra-oral radiographs were taken at re-evaluation visits. Radiographic measurements had been made by trained postdoctoral student, MY. Duplicate measurements were taken on radiographs of 15 patients on different days [9]. The mean difference of the two sets of measurements was 0.13 mm ± 0.09 mm, the Pearson correlation was 0.97. Marginal bone loss (MBL) was defined as the mean of mesial and distal bone resorption measured by digital caliper (Niigata Seiki, Niigata, Japan). Distance between the fixture/abutment junction and the marginal bone level at both mesial and distal sites of measured implants on dental and/or panoramic radiographs was recorded. Degree of bone resorption was obtained from the enlargement ratio based on the length of the fixture, as described [13]. The baseline was defined as the time of implant placement.

### 2.5. Definition of Implant Failure and Success

Implant failure was classified into four categories: loss of implants due to LoO or peri-implantitis (PI (L)); presence of peri-implantitis (PI); and MBL ≥3 mm without inflammatory reaction, with reference to the latest classification [12]. Peri-implantitis was defined as MBL ≥3 mm with a peri-implant pocket depth of >5 mm with BOP and/or suppuration [6,14]. Success of implant treatment was defined as MBL <3 mm. Survival time was defined as time from implant insertion to removal or last follow-up [15]. Implant outcome variables, including MBL and BOP, were recorded at follow-up. Pocket depth measurements may be unreliable because of the difficulty of probing for implants compared to natural teeth, however marginal bone loss can be evaluated by dental and panoramic X-ray imaging. This is the major reason we did not include peri-implant mucositis in this study.

### 2.6. Statistical Analyses

Results were analyzed using SPSS version 28 (IBM, Chicago, IL, USA) with *p* < 0.05 deemed significant. Data were evaluated using ANOVA, chi-squared test, Fisher’s exact test, Welch’s test or Student’s *t*-test, as appropriate. Chi-squared test, Welch’s test, Fisher’s exact test and ANOVA were used to compare related factors among Stage and Grade of periodontitis. Principal components analysis (PCA) and two-step cluster analysis (CA) were performed to identify similarities and differences among observed variables at implant and patient level. PCA examined 15 variables (age, gender, implant brand, function duration, implantation position, number of implants, Stage, Grade, BA, single implant, terminal molar, type of opposite tooth, parafunction, smoking and diabetes). We selected principal components with eigenvalues >1. CA was performed based on the principal component scores. The number of clusters was determined based on Akaike information criterion (AIC) [16].

## 3. Results

### 3.1. Study Participants

Of the 98 (383 implants) patients who had received comprehensive treatment for periodontitis between 2006 and 2018 at Ohu University Dental Hospital (Fukushima, Japan), 84 patients (325 implants) were enrolled for this study. For the aforementioned reasons, 14 patients with 58 implants had to be excluded. Data on the 84 enrolled patients and their 325 implants are presented in Table 1. Differences in prevalence of peri-implant disease were assessed at both patient and implant level. There was no significant difference in patient age by Stage or Grade at first visit. Mean follow-up period was significantly longer in patients implanted with POI than in those implanted with Bmk, regardless of Stage or Grade (*p* < 0.01).

The distribution between maxilla and mandible did not differ significantly among the two groups, although a slightly greater proportion of implants was observed in posterior mandibles. Implant lengths were almost all within the 8–12 mm range, with 10 mm accounting for 90.5% of them. Implant-supported reconstructions included 50 single units and 275 multi-units fixed dentures.

### 3.2. Treatment Outcomes

#### 3.2.1. Patient Level

(1) Survival and success rates

Survival and success rates were 85.7% and 72.6%, respectively. Of the 84 patients, 4 (4.8%) experienced loss of implants due to peri-implantitis, 7 (8.3%) showed peri-implantitis and 7 showed MBL ≥ 3 mm, 61 (72.6%) showed MBL < 3 mm and 5 (7.1%) showed LoO (Table 2A).

(2) Treatment outcomes

Instances of implant loss due to peri-implantitis and MBL ≥3 mm were significantly higher in Stage IV patients (*p* < 0.05). Instances of MBL < 3 mm were significantly lower in Stage IV patients (*p* < 0.05) (Table 3, Appendix A). Mean MBL differed between smokers (2.19 ± 0.82 mm) and non-smokers (1.57 ± 0.62 mm) (*p* < 0.01; Student’s *t*-test). The number of implants and degree of bone resorption were significantly higher in the BA than non-BA group (*p* < 0.05, *t*-test). Mean MBL in BA and non-BA were 4.83 ± 3.46 mm and 3.33 ± 2.63 mm, respectively.

(3) Relationship between periodontitis severity and number of implants

The mean numbers of implants placed in Stage IV, III and I patients were 5.71 ± 3.52, 3.24 ± 2.36 and 3.07 ± 1.98, respectively. The number in Stage IV patients was significantly higher than in Stage I and III patients (*p* < 0.05; ANOVA) and in Grade C (4.50 ± 3.18) than in Grade A (2.50 ± 1.37), (*p* < 0.05; Welch’s test).

(4) Threshold for increased risk of implant failure by number of implants

The prevalence of peri-implant disease differed by number of implants per patient (*p* < 0.01) at the rate of MBL ≥3 mm with three or more implants and peri-implantitis with eight or more implants, respectively.

(5) PCA and CA

The results shown in Figure 1A were obtained by PCA. In the subsequent analysis, five principal components with eigenvalues ≥1 were assessed. The % variance of component 1 was 21.3%. Component 1 was strongly influenced by Stage and Grade. Factors other than those were functional duration, number of implants and gender. See Figure 1A for details of the 2nd to 5th principal components.

Four clusters were revealed by CA (Table 4). The number of clusters was determined based on Akaike information criterion (AIC: 282.7).

Subgroups included difficult-case patients with several implants and a high proportion of BA (cluster 1), with a high number of non-compliers with smoking (cluster 4), and compromised hosts with poor prognosis despite few conditions considered high-risk (cluster 2) for developing peri-implant disease. In contrast, there was a low-risk group (cluster 3) with low severity of periodontitis and better treatment outcomes.

#### 3.2.2. LoO

The LoO group had different characteristics from other peri-implant disease groups, and there was effect on neither severity of periodontitis nor number of implants. In contrast, BA combination group (*p* < 0.05), POI group (*p* < 0.01) and single implant prosthesis (*p* < 0.01) were highly significant factors of implant failure (Table 3). Occlusal functional duration of implant was significantly shorter in the LoO group than in the group with MBL < 3 mm (*p* < 0.05; ANOVA).

#### 3.2.3. Implant Level

(1) Survival and success rates

Survival and success rates were 96.3% and 87.1%, respectively. Of the 325 implants, 5 (1.5%) dropped out due to peri-implantitis, 14 (4.3%) suffered from peri-implantitis, 16 (4.9%) showed MBL ≥3 mm, 283 (87.1%) showed MBL <3 mm and 7 (2.2%) dropped out due to LoO (Table 2B).

(2) Treatment outcomes

Stage IV (*p* < 0.05) and BA (*p* < 0.05) were significantly higher in PI (L) group. Stage IV (*p* < 0.05), implant brand (Bmk, *p* < 0.01) and smoking (*p* < 0.01) were significantly higher in PI group. Stage IV (*p* < 0.05) was significantly higher in MBL ≥3 mm group. BA (*p* < 0.05), implant brand (POI, *p* < 0.01), smoking (*p* < 0.01) and single implant (*p* < 0.01) were significantly higher, while functional duration was significantly shorter (*p* < 0.05) in LoO group (Table 3, Appendix A). In contrast, Stage IV (*p* < 0.05), single implant (*p* < 0.01) and smoking (*p* < 0.01) were significantly lower in MBL <3 mm group. Mean MBL differed between smokers (2.44 ± 1.51 mm) and non-smokers (1.66 ± 0.90 mm) (*p* < 0.01; Welch’s test). The proportion of patients that received guided bone regeneration was significantly higher in Stage III and in Grade C groups (*p* < 0.01; chi-square test). Mean MBL differed between Stage I (1.52 ± 0.67 mm) and IV groups (1.92 ± 1.16 mm) (*p* < 0.05; Welch’s test).

(3) PCA and CA

The results shown in Figure 1B were obtained by PCA. In the subsequent analysis, six principal components with eigenvalues ≥ 1 were used. The % variance of component 1 was 17.1%. Component 1 was strongly influenced by Stage and Grade. Factors other than that were implant brand, functional duration, opposing tooth and gender. See Figure 1B for details of the 2nd to 6th principal components.

Three clusters were revealed by CA (Table 5). The number of clusters was determined based on Akaike information criterion (AIC: 1133.7). Subgroups included difficult-case patients with several implants and a high proportion of BA (cluster 2), with a high number of non-compliers with smoking (cluster 3). In contrast, there was a low-risk group (cluster 1) with low severity of periodontitis and better treatment outcomes.

## 4. Discussion

Within the limitation of this retrospective study, it tends to show that severity (Stage) of periodontitis is the most critical determinant of increased risk of peri-implant diseases. In contrast, implant loss by LoO occurs regardless of periodontitis severity, suggesting that occlusal overload, parafunction and BA may be involved in the pathogenesis. Implant failure may result from single, combined or multiple factors. It is important to perform patient risk assessment and to establish a suitable preventive regimen for each patient as personalized medicine.

Our finding that patients with severe periodontitis tend to suffer more from peri-implant disease, including peri-implantitis, was similar to findings reported previously [17,18,19]. Kang et al. reported early failure before occlusal function [20]. Low incidence of early failure in the present study suggests surgical skill within our team was sufficient.

The number of patients receiving BA and the number of implants were significantly higher in Stages III, IV and Grade C groups. Possible explanations include fewer remaining teeth, more hopeless teeth and greater bone resorption in Stage III and IV patients. Although BA is an established practice, it is not risk-free, as flap dehiscence and post-operative infection occasionally occur [21]. In addition, artificial bone such as hydroxyapatite or β-TCP may be of poorer quality than real bone [22] and osseointegration between an implant and artificial bone may be weaker than that between an implant and autologous bone.

Canullo et al. reported a higher prevalence of surgically-triggered peri-implantitis than prosthetically- and plaque-induced peri-implantitis, which supports the possibility that the surgical skill of the dentist is a prime factor in implant failure [6]. In fact, the success rate of implant surgery was not 100% in our clinics; if treatment modalities such as BA are involved, the challenge to dentists may increase and treatment success rate drop.

Blood flow around an implant is inferior to that around a natural tooth and may be obstructed, causing increased risk of destruction of hard and soft tissues in patients with multiple implants [23]. In addition, subtle errors during surgery may increase the probability of prosthetically- and surgically-triggered peri-implantitis [6]. If the interface is not tight, a slight gap may be present between the implant and abutment, however even when no gap was observed on dental X-ray radiographs, bacteria were found between the implant and abutment during maintenance [24]. It was reported that implant-abutment assemblies with less tight interfaces lead to inflammatory reactions due to bacterial infection [25].

Papantonopoulos et al. reported cluster patterns and classified patients into two phenotypes: “resistance” and “susceptibility” to peri-implantitis [26]. Likewise, we found similar clusters for peri-implantitis, with a group of clusters characterizing possible risk factors. Therefore, risk assessment of periodontitis in each patient is useful to predict implant failure.

At the patient level, clusters 1, 2 and 4 were a group of patients with poorer outcomes than cluster 3, and included possible risk factors such as high periodontitis severity, BA, smoking and increased number of implants. Similarly, at the implant level, clusters 2 and 3 included possible risk factors that may increase treatment difficulty, such as high periodontitis severity, smoking and opposing teeth.

The worst failure of implant treatment is loss of the implant body, which can be attributed to peri-implantitis, LoO and damage to the implant body [2,27]. In the present study, although the final outcomes of peri-implantitis and LoO were the same, the clinical features of LoO distinctly differed from the former. Time to onset was significantly shorter in LoO group than other groups, and no association with severity of periodontitis was observed. All patients with implant-body detachment due to peri-implantitis were found in Stage IV/Grade C, while LoO was also found in Stage I and Grade A. Occlusal overload was reported to be involved in LoO [28]. Compared to natural teeth, implants without periodontal ligaments may lack osseointegration and detach in a short period of time following excessive occlusal force.

The present study has some limitations. Multivariate analysis to calculate odds ratios requires a sufficient number of samples per factor [5,13], however the number of samples in this study might have been insufficient to calculate the odds ratios robustly. In addition, baseline MBL measurements were not consistently obtained at the same times as previous studies, including at time of implant placement [29], at 1-year after occlusal function [5] and at 1-year after final prosthesis [13]. It was reported that peri-implant bone had been remodeled after implant placement and occlusal function [29], however the observation period in this study was relatively shorter than those in other studies. Accordingly, we took MBL before bone remodeling as the baseline and defined pathological bone resorption as ≥3 mm.

There has been wide heterogeneity in the prevalence of peri-implantitis among previous reports [30,31]. Smoking, history of periodontitis, width of keratinized mucosa and systemic diseases including diabetes mellitus and hypertension may influence implant survival and success [32]. Our results regarding smoking were similar to those of a previous study [13], although this factor might have had minimal impact on implant failure in our study, as relatively few smokers were included (as shown in Table 1). History of periodontal disease has been reported as a possible risk factor for peri-implantitis [13,33], however, in the absence of clinical records, self-reports by patients relying on their memories may be unreliable. This is a main reason why researchers should use Stage and Grade instead of history of periodontal disease.

## 5. Conclusions

This study demonstrates that severity of periodontitis is a significant risk for development of peri-implant disease at both patient and implant levels. The Stage and Grade classification of periodontitis could be a useful tool to predict the treatment outcome of implant therapy of patients with different types of periodontitis. Further study is required to compare treatment outcome between high- and low-risk groups for peri-implant diseases in a prospective cohort study.

## Figures and Tables

**Figure 1 jpm-12-01723-f001:**
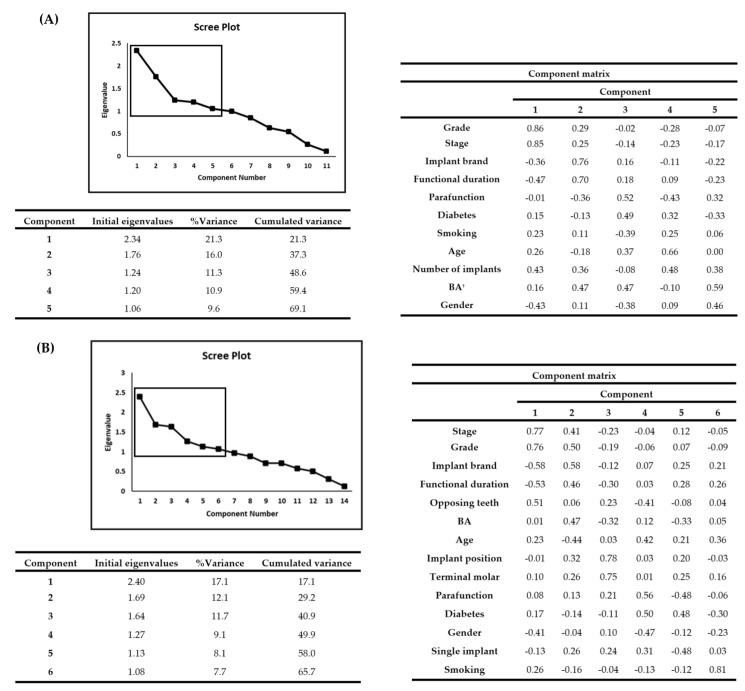
(**A**). Parameters on the steep slope of the scree plot are typically selected as principal. Principal components analysis at patient level. As a result of principal components analysis, the 5 components that showed eigenvalues >1 were selected from the 11 evaluated. †: bone augmentation (**B**). Principal components analysis at implant level. Of the 14 evaluated, 6 components that showed eigenvalues >1 were selected.

**Table 1 jpm-12-01723-t001:** Participant and implant characteristic features.

Features at the Patient Level (*n* = 84)		Findings	Features at the Implant Level (*n* = 325)		Findings
Gender	Male	41.7%	Severity of periodontitis (Stage)	Ⅰ	13.2%
	Female	58.3%		Ⅱ	16.0%
Age (Years)		54.7 ± 11.5		Ⅲ	33.8%
	≤49	28.6%		Ⅳ	36.9%
	50–59	33.3%	Risk of periodontitis (Grade)	A	12.3%
	60–69	29.8%		B	24.0%
	70–79	8.3%		C	63.7%
	≥80	0.0%	Implant brand	POI	48.6%
Diabetes	HbA1c ≤ 7%	6.0%		Bmk	51.4%
Smokers ^†^		8.3%	Implant diameter (mm)		3.3–5.0
	≥10 cigarettes/day	1.7%	Implant length (mm)		7.0–11.5
Severity of periodontitis (Stage)	Ⅰ	16.7%	Functional duration (year)		5.2 ± 2.5
	Ⅱ	17.9%		POI	6.7 ± 2.2
	Ⅲ	40.5%		Bmk	3.8 ± 1.8
	Ⅳ	25.0%	Type of prosthesis	Single-unit	21.5%
Age of patients among four groups (Stage)		Ⅰ; 51.1 ± 14.7 (29–74)		Multi-unit fixed	78.5%
		Ⅱ; 59.1 ± 7.5 (48–71)	Implant position		
		Ⅲ; 54.1 ± 11.0 (31–74)	Upper jaw	incisor	7.1%
		Ⅳ; 54.8 ± 11.9 (26–69)		canine	1.8%
Risk of periodontitis (Grade)	A	19.0%		premolar	15.1%
	B	26.2%		molar	26.5%
	C	54.8%	Lower jaw	incisor	2.8%
Age of patients among three groups (Grade)		A; 50.9 ± 13.2 (29–74)		canine	2.5%
		B; 59.7 ± 10.0 (31–74)		premolar	10.8%
		C; 53.6 ± 10.9 (26–70)		molar	33.5%
Implant brand	POI	52.4%	Opposing teeth	natural teeth	58.5%
	Bmk ^‡^	47.6%		implant	41.5%
Functional duration (year)		5.1 ± 2.5	Terminal molar		37.5%
	POI	6.6 ± 2.1	Bone augmentation		21.5%
	Bmk	3.4 ± 1.7	Parafunction		73.5%
Number of implants		3.87 ± 3.02			
	1~3	59.5%	Survival rate		96.3%
	4~6	22.6%	Success rate		87.1%
	7~9	14.3%			
	≥10	3.6%			
Bone augmentation		35.7%			
Parafunction		79.8%			
Survival rate		85.7%			
Success rate		72.6%			

^†^: Ex-smokers were counted as non-smokers. ^‡^: Brånemark system.

**Table 2 jpm-12-01723-t002:** Distribution of treatment outcomes classified by Stage and Grade. (**A**). Patient level ^†^ (*n* = 84). (**B**). Implant level (*n* = 325).

(A)						
			MBL ^¶^			
	PI (L) ^‡^	PI ^§^	≥3 mm	<3 mm	LoO ^††^	SUM
Stage Ⅰ	0	0	1	11	2	14
Ⅱ	0	1	1	13	0	15
Ⅲ	0	3	1	27	3	34
Ⅳ	4	3	4	10	0	21
Grade A	0	0	0	14	2	16
B	0	3	2	17	0	22
C	4	4	5	30	3	46
**(B)**						
			**MBL**			
	**PI (L)**	**PI**	**≥3 mm**	**<3 mm**	**LoO**	**SUM**
Stage Ⅰ	0	0	2	39	2	43
Ⅱ	0	2	2	48	0	52
Ⅲ	0	3	2	101	4	110
Ⅳ	5	9	10	95	1	120
Grade A	0	0	1	37	2	40
B	0	5	3	70	0	78
C	5	9	12	176	5	207

^†^: Patient level shows the worst outcomes in implant treatment per patient. ^‡^: Peri-implantitis (lost). ^§^: Peri-implantitis (surviving). ^¶^: Marginal bone loss, MBL (<3 mm) indicates success of implant treatment. ^††^: Lack of osseointegration.

**Table 3 jpm-12-01723-t003:** Factors related to implant failure (peri-implant disease, lack of osseointegration).

Treatment Outcome		PI (L)	PI	MBL		LoO
				≥3 mm	<3 mm	
Patient Level						
	Stage Ⅳ ^†^	*		*	*	
Implant Level						
	Stage Ⅳ ^†^	*	*	*	*	
	GBR ^†^	*				*
	Implant brand (POI) ^†^					**
	Implant brand (Bmk) ^†^		**			
	Smoker ^†^		**			**
	Functional duration ^‡^					*
	Type of prosthesis single ^†^				**	**

*: *p* < 0.05. **: *p* < 0.01. ^†^: Fisher’s exact test. ^‡^: One-way ANOVA.

**Table 4 jpm-12-01723-t004:** Features among four clusters at patient level.

		Cluster 1		Cluster 2		Cluster 3		Cluster 4	
		(*n* = 22), %		(*n* = 35), %		(*n* = 15), %		(*n* = 12), %	
Age (mean)		63.3		50.2		48.9		59.3	
Gender	Male	10	45.5	19	54.3	2	13.3	4	33.3
	Female	12	54.5	16	45.7	13	86.7	8	66.7
Implant brand	POI	8	36.4	23	65.7	7	46.7	6	50
	Bmk	14	63.6	12	34.3	8	53.3	6	50
Stage	Ⅰ	3	13.6	0	0	10	66.7	1	8.3
	Ⅱ	4	18.2	3	8.6	4	26.7	4	33.3
	Ⅲ	12	54.5	18	51.4	1	6.7	3	25
	Ⅳ	3	13.6	14	40	0	0	4	33.3
Grade	A	2	9.1	0	0	12	80	2	16.7
	B	6	27.3	8	22.9	3	20	5	41.7
	C	14	63.6	27	77.1	0	0	5	41.7
Number of implants (mean)		5		3		2		6	
Treatment outcome	PI (L)	1	4.5	3	8.6	0	0	0	0
	PI	2	9.1	4	11.4	0	0	1	8.3
	MBL ≥ 3 mm	2	9.1	4	11.4	0	0	1	8.3
	MBL < 3 mm	15	68.2	23	65.7	14	93.3	9	75
	LoO	2	9.1	1	2.9	1	6.7	1	8.3
Marginal bone loss (mm/mean)		1.62		1.71		1.31		1.75	
Functional duration (year/mean)		4.2		5.6		5.7		4.7	
Bone augmentation	No	5	22.7	24	68.6	13	86.7	12	100
	Yes	17	77.3	11	31.4	2	13.3	0	0
Parafunction	No	1	4.5	6	17.1	0	0	10	83.3
	Yes	21	95.5	29	82.9	15	100	2	16.7
Smoking	No	21	95.5	33	94.3	15	100	8	66.7
	Yes	1	4.5	2	5.7	0	0	4	33.3
Diabetes	No	17	77.3	35	100	15	100	12	100
	Yes	5	22.7	0	0	0	0	0	0

**Table 5 jpm-12-01723-t005:** Features among three clusters at implant level.

		Cluster 1		Cluster 2		Cluster 3	
		(*n* = 148), %		(*n* = 101), %		(*n* = 76), %	
Age (mean)		59.4		54		61.4	
Gender	Male	56	47.9	57	32.9	19	54.3
	Female	61	52.1	116	67.1	16	45.7
Implant brand	POI	86	73.5	59	34.1	13	37.1
	Bmk	31	26.5	114	65.9	22	62.9
Implant position	incisor	14	12	11	6.4	7	20
	canine	5	4.3	8	4.6	1	2.9
	premolar	30	25.6	45	26	9	25.7
	molar	68	58.1	109	63	18	51.4
Stage	Ⅰ	43	36.8	0	0	0	0
	Ⅱ	17	14.5	24	13.9	11	31.4
	Ⅲ	40	34.2	65	37.6	5	14.3
	Ⅳ	17	14.5	84	48.6	19	54.3
Grade	A	40	34.2	0	0	0	0
	B	35	29.9	32	18.5	11	31.4
	C	42	35.9	141	81.5	24	68.6
Treatment outcome	PI(L)	0	0	4	2.3	1	2.9
	PI	1	0.9	8	4.6	5	14.3
	MBL ≥ 3 mm	3	2.6	11	6.4	2	5.7
	MBL < 3 mm	110	94	149	86.1	24	68.6
	LoO	3	2.6	1	0.6	3	8.6
Marginal bone loss (mm/mean)		1.48		1.77		2.44	
Functional duration (year/mean)		6.6		4.5		4.5	
Bone augmentation	No	95	81.2	131	75.7	29	82.9
	Yes	22	18.8	42	24.3	6	17.1
Type of prosthesis (single-unit)	No	92	78.6	153	88.4	30	85.7
	Yes	25	21.4	20	11.6	5	14.3
Terminal molar	No	77	65.8	105	60.7	21	60
	Yes	40	34.2	68	39.3	14	40
Opposing teeth	natural teeth	105	89.7	71	41	14	40
	implant	12	10.3	102	59	21	60
Parafunction	No	33	28.2	42	24.3	11	31.4
	Yes	84	71.8	131	75.7	24	68.6
Smoking	No	117	100	173	100	0	0
	Yes	0	0	0	0	35	100
Diabetes	No	98	83.8	173	100	35	100
	Yes	19	16.2	0	0	0	0

## Data Availability

The data presented in this study are available on request from the corresponding author.

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
