# Peer review of "The Stages and Grades of Periodontitis Are Risk Indicators for Peri-Implant Diseases—A Long-Term Retrospective Study"

_jpm, 2022, doi:10.3390/jpm12101723_

Round 1
Reviewer 1 Report
Dear authors,
I consider the article entitled “The Stages and Grades of Periodontitis are Risk Indicators for Peri-implant Diseases” very interesting and very useful for the dental practice.
In Introduction the authors wrote:
Peri-implantitis is a multifactorial disease [4, 6]. I consider that it is very important for the practitioners to present in detail all the factors involved in the etiology of this disease.
Also, the autors wrote “Based on the 2017 World Workshop on the Classification of Periodontal and Periimplant Diseases and Conditions [8, 9], periodontitis was newly sub-classified by Stage and Grade”. This aspect must be detailed.
2. Materials and Methods
Study design
The research protocol was approved by Ethics Board, Ohu University (reference number 222). Must be mentioned the year
The same problem is encountered in: Institutional Review Board Statement: The research protocol was approved by Ethics Board, Ohu University (reference number 222).
Author Response
Dear Reviewer 1
Thank you for the thoughtful and constructive feedback you provided regarding our manuscript, “The Stages and Grades of Periodontitis are Risk Indicators for Peri-implant Diseases”. We hope our revision will be acceptable for your constructive comments.
Keiso Takahashi DDS, PhD, Professor
Division of Periodontics, Department of Conservative Dentistry
Ohu University School of Dentistry
31-1, Misumido, Tomitamachi, Koriyama,
Fukushima 963-8611
Japan
In Introduction the authors wrote:
Peri-implantitis is a multifactorial disease [4, 6]. I consider that it is very important for the practitioners to present in detail all the factors involved in the etiology of this disease.
--> Thank you for your suggestion. There are many candidates for risk indicators/risk factors for peri-implant diseases and controversy still exists for the possible factors because of wide heterogeneity of clinical features of patients and operators, such as the severity of periodontitis, age, systemic diseases, smoking habits, dentist’s skill, etc. Therefore, it is quite difficult to present the all factors and then we have cited previous papers (5, 9).
Also, the authors wrote “Based on the 2017 World Workshop on the Classification of Periodontal and Peri-implant Diseases and Conditions [8, 9], periodontitis was newly sub-classified by Stage and Grade”. This aspect must be detailed.
--> Thank you for your suggestion. The details of new classification had been already reported in ref. 4 and 12 and these concepts are now well recognized. Therefore, we have omitted additional explanation for the new classification.
- Materials and Methods
Study design
The research protocol was approved by Ethics Board, Ohu University (reference number 222). Must be mentioned the year
The same problem is encountered in: Institutional Review Board Statement: The research protocol was approved by Ethics Board, Ohu University (reference number 222).
--> Thank you for your suggestion. We have added the year (2018).
Reviewer 2 Report
Revisions to the article entitled “The Stages and Grades of Periodontitis are Risk Indicators for 2 Peri-implant Diseases” By Mikiko Yamazaki 1, Kosaku Yamazaki 2, Yuh Baba3, Hiroshi Ito4, Bruno G Loos5 and Keiso Takahashi 2, *
Briefly, this work seems to have aimed at investigating the relationship between peri-implant disease and the severity and complexity of periodontal disease.
The results show different profile (cluster) with high or low risk of peri-implant disease. The strength of this study is the method to determine the clusters. It has the advantage of exploring a field where there are still many risk factors to be confirmed in terms of weight/impact in the development of peri-implant disease..
General concept comments
The weaknesses lie in:
The lack of definition of the primary and secondary objectives of the study.
Too many factors (15) are investigated in this retrospective study
Non-calibration of the unique examiner.
Very confusing presentation of results.
A non-updated bibliography on the subject.
Too many many off-topic issues in the discussion
General remarks
Since this is the scope of the Journal of Personalized Medicine, I suggest that it would be more relevant to present only the results at the patient level.
There is much confusion in the use of the terms implant success and implant failure. Choose to always use the same term and present your results according to the term you have chosen. I would advise you to speak about implant failure and not implant surviving.
The Chicago classification is not new as it was published in June 2018, remove the term new throughout the text.
Specific comments to authors
The TITLE should indicate that it is a retrospective study.
ABSTRACT: It should be rewritten after clarifying the primary and secondary objectives of this study and thus modifying the presentation and analysis of the results. It is the article “in miniature”.
|
Page
|
Line |
Comments |
|
1 |
17 |
Objectives used to be the last sentence of the introduction, the aim of the study |
|
1 |
30 |
Delete new . Add to key words prevalence- risk indicators/risk factors- |
INTRODUCTION
The introduction needs to be rewritten with relevant and updated bibliographic references.
The definition of implant success and implant survival should be given with a bibliographic reference. “Implant survival rate or positive outcome was defined as the absence of progressive bone loss as evaluated on radiographs after 6 months, at one year and once yearly afterwards.” Berglundh et al. (2) 2018;
|
Page
|
Line |
Comments |
|
1 |
36 |
Delete “recently” |
|
1 |
42 |
Delete “with surgical skills of dentists being the most significant factor” Cite Heitz-Mayfield LJA, Heitz F, Lang NP. Implant Disease Risk Assessment IDRA-a tool for preventing peri-implant disease. Clin Oral Implants Res. 2020 Apr;31(4):397-403. doi: 10.1111/clr.13585. Epub 2020 Feb 20. PMID: 32003037. Sarbacher A, Papalou I, Vagia P, Tenenbaum H, Huck O, Davideau JL. Comparison of Two Risk Assessment Scores in Predicting Peri-Implantitis Occurrence during Implant Maintenance in Patients Treated for Periodontal Diseases: A Long-Term Retrospective Study. J Clin Med. 2022 Mar 20;11(6):1720. doi: 10.3390/jcm11061720. PMID: 35330046; PMCID: PMC8948905.
|
|
2 |
45 |
Please begin the introduction by this sentence.
|
|
2 |
49-51 |
To put this sentence if necessary, in the DISCUSSION it is the "justification of the method used in this study" |
|
2 |
53-56 |
Delete “newly” and “new” |
|
2 |
57-60 |
The last sentence should be the aim revised of the study. There are too many aims in this study please reduce to one or two aims maximum. For example “the aim of this study was to evaluate the factors of implant failure in periodontal patients and their impact on the prognosis of having a peri-implant disease/failure.”or to evaluate the relationship between implant failure and stage and grade ? Please choose one primary outcome and secondary outcomes. |
Material and Methods
The material and method should be restructured and should not include results.
|
Page
|
Line |
Comments |
|
2 |
62 |
Write that it is a retrospective study
|
|
2 |
66 |
Delete “84 patients (325 implants) were enrolled for this study. « these are RESULTS |
|
2 |
67 |
Here please add the inclusion criteria referred to L.70-81 The inclusion criteria should be listed.
|
|
2 |
68-69-70 |
Delete the number of patient between () |
|
2 |
84 |
Please give the results of the intra-individual reliability test of the “trained post-doctoral student” |
|
2-3 |
95-99 |
Remove the parameter concerning the BA there is too much disparity in the bone regeneration surgical procedures. it is a huge bias. This parameter should not be analysed. |
|
3 |
104 |
to put this table in the results chapter In “Diabetes” , specify glycated haemoglobin (HbA1c) less or equal to 7 In smokers, specify if the number of cigarettes smoked is < or > 10 cigarettes/day.. Because these are 2 grade modifiers that you used when you classified your patients' periodontitis.
|
|
3 |
111 |
“Follow-up examinations were conducted by experienced periodontists. “ is different from “"All reported measurements were made during 2018-2020 by a trained postdoctoral student (MY). " please explain the method. |
|
3 |
116 |
Delete “as either positive or negative.” |
|
4 |
118 |
Define the method of calibrating the radiographs to reproducibly measure the bone level.
|
|
4 |
132-133 |
Complete the definition of Implant survival rate Or provide a reference of “Success of implant treatment was defined as MBL˂3 mm” Please give the definition of implant survival with reference.
|
Results
Please present the patients and implants data in two separated tables based upon table 1.
Then after, present the result concerning the primary outcome.
Then present the results concerning the secondary outcomes.
|
Page
|
Line |
Comments |
|
4 |
153 |
« Mean follow-up period » which is ? give the number . |
|
4 |
157 |
What are these « two groups, » ? |
|
4 |
159-160 |
Please delete these sentence “Implant-supported reconstructions included 50 single crowns, 246 fixed partial dentures, and 29 overdentures.” |
|
4 |
162 |
You choose to present the results concerning treatment outcomes in 2 different chapter based upon 2 levels : patient and implant However in table 2 B the results showed "implant level". Please make 2 different table for each chapter. idem for table 3 all the data are mixed |
|
4 |
165 |
Add to table 1 “Survival and success rates were 85.7% and 72.6%, respectively”. |
|
5 |
Table 2 170 |
Give the total number of patients |
|
5 |
Table 2 171-172-173 |
present it differently patient who has lost an implant patient with peri-implantitis (specify if treated and if so how) |
|
5 |
176 |
≧
|
|
5 |
178-179 |
“Mean MBL differed between smokers (2.19 ± 0.82 mm) and non-s…/... test).” This is not the aim of this study. Otherwise, it should be defined in the material and method and in the objective of the study that you will compare MBL between smokers and non-smokers. Delete. |
|
5 |
Table 3 181 |
Add to the title of the table “compared to stage III/II/I independently” |
|
5 |
Table 3 181 |
The results of the other stages and grades should also be given |
|
5 |
186-190 |
Please note that these data correspond to the definition of a stage IV periodontitis case, so they do not provide relevant information. Delete |
|
5 |
192-195 |
it is not the aim of the study. Delete. Is there more BA in elevated periodontitis stage? Is there a relationship between stage and MBL ? |
|
5 |
203 |
« F ive principal components » which are ? |
|
6 |
207-211 |
this type of presentation of results is very difficult for the reader to understand. please simplify the text. |
|
6 |
Figure 1 213-234 |
One number per figure and one number per table don’t use “ A or B”. Simplify the figures and explain why 11 or 14? |
|
7 |
251-253 |
Occlusal functional duration of implant was significantly shorter in the LoO group than the group with 252 MBL<3 mm (p<0.05; ANOVA). Please delete this is not the aim of the study. |
|
7-8 |
262-294 |
Rewrite with precise separation of patient level data from implant level data. I suggest that it would be more relevant to present only the results at the patient level.
|
Discussion
The discussion should be re-structured according to the usual plan which is reminding the reader of the main result of your study and compare it to other data in the literature.
Justify your methodology and point out its limitations, biases and risks of error.
Compare each result with those in the literature.
Starting with the primary outcome and then the secondary outcomes. Then try to give an explanation of the observed phenomena supported by bibliographic references that you have to improve.
You have to cite more relevant bibliographical references (related to your topic) and you should add those I recommend and others you could find.
|
Page
|
Line |
Comments |
|
8 |
296 |
Replace by "within the limits of this retrospective study it tends to show that ...." |
|
8 |
297-298 |
Please delete these sentence. “In contrast, implant loss by LoO occurred regardless of periodontitis severity, suggesting that occlusal over load, parafunction, and BA, may be involved in the pathogenesis.” |
|
8 |
299-301 |
“Implant failure may result from single, combined, or multiple factors. It is important to perform patient risk 300 assessment, and to establish a suitable preventive regimen for each patient as personalized medicine. » This should be part of the introduction |
|
8-9-10 |
304 327 348 350 |
There should be no results called (table or figures) in the discussion. Delete all of them. |
|
8 |
305-307 |
« Chrcanovic et al. reported early failure before occlusal function [14]. Low incidence of early failure in the present study suggests surgical skill in our team was sufficient.” Please delete it is out of scope. |
|
8 |
311-315 |
Please delete it is out of scope. |
|
8 |
319-320 |
« implant failure. » or peri-implant disease ? |
|
9 |
323 |
« Surgical skill of dentist » It is not the aim of the study. Delete At the beginning of the discussion write a sentence to justify your method which consisted in an only one experimented periodontist who performed all the surgical procedures in all the 84 patients |
|
9 |
323-325 |
“In fact, the success rate of implant surgery was not 100% in our clinics, if treatment modalities such as BA are involved, the challenge to dentists may increase and treatment success rate drop.” It is not the aim of the study. At the beginning of the discussion write a sentence to justify your method which consisted in an only one experimented periodontist who performed all the surgical procedures in all the 84 patients. Delete |
|
9-10 |
337-342 |
“In addition, subtle errors during surgery may increase the probability of prosthetically and surgically triggered peri-implantitis [5]. If the interface is not tight, a slight gap may be present between the implant and abutment, but even when no gap was observed on dental X-ray radiographs, bacteria was found between the implant and abutment during maintenance [19]. It was reported that implant-abutment assemblies with less tight interfaces led to inflammatory reactions due to bacterial infection [20].” Please delete it is out of scope. |
|
10 |
343 |
“The biggest failure of implant treatment is detachment of the implant body,” poorly said please rephrase |
|
10 |
344 |
“damage to the implant body” please delete |
|
10 |
344-347 |
In the present study, although the final outcomes of peri-implantitis and LoO were the same, the clinical features of LoO distinctly differed from the former. Functional duration was sig nificantly shorter in LoO. poorly said please rephrase |
|
10 |
350-352 |
Compared to natural teeth, implants without periodontal ligaments may lack osseointegration and detach in a short period of time following excessive occlusal force. please delete |
|
10 |
359-360 |
« the observation period » please give the number or mean number. |
|
10 |
371-374 |
These sentences must be placed in the introduction |
|
10 |
375-377 |
“In practice, pocket depth measurements may be unreliable because of the difficulty of probing for implants compared to natural teeth, however, marginal bone loss could be evaluated by dental and panoramic X-ray” This sentence must be placed in the material and methods section. Is it reproducible ? |
|
10 |
|
« This is the major reason we did not include peri-implant mucositis in this study. » Add to the non-inclusion criteria in the material and methods section |
Conclusion :
|
Page
|
Line |
Comments |
|
10 |
381 |
“Peri-implant diseases have heterogeneous features caused by multiple factors” must be placed in the introduction and delete here. |
Conclude on the most relevant and interesting result of your work and the prospects for applying your results or for further studies. This should not be a summary or a repetition of the results. It should not be longer than 2 to 3 sentences.
Add and use these references
1. Arunyanak, SP, Sophon, N, Tangsathian, T, Supanimitkul, K, Suwanwichit, T, Kungsadalpipob, K. The effect of factors related to periodontal status toward peri-implantitis. Clin Oral Impl Res. 2019; 30: 791– 799. https://doi.org/10.1111/clr.13461
2. Berglundh T, Armitage G, Araujo MG, Avila-Ortiz G, Blanco J, Camargo PM, et al. Peri-implant diseases and conditions: Consensus report of workgroup 4 of the 2017 World Workshop on the Classification of Periodontal and Peri-Implant Diseases and Conditions. J Clin Periodontol 2018;45:S286-S291
3. Heitz-Mayfield LJA, Heitz F, Lang NP. Implant Disease Risk Assessment IDRA-a tool for preventing peri-implant disease. Clin Oral Implants Res. 2020 Apr;31(4):397-403. doi: 10.1111/clr.13585. Epub 2020 Feb 20. PMID: 32003037.
4. Hu C, Lang NP, Ong MM, Lim LP, Tan WC. Influence of periodontal maintenance and periodontitis susceptibility on implant success: A 5-year retrospective cohort on moderately rough surfaced implants. Clin Oral Implants Res. 2020 Aug;31(8):727-736. doi: 10.1111/clr.13621. Epub 2020 Jul 6. PMID: 32459865.
5. Rakic M, Galindo-Moreno P, Monje A, et al. How frequent does peri-implantitis occur? A systematic review and meta-analysis. Clin Oral Investig. 2018;22:1805-1816.
6. Schwarz F,Derks J,Monje A,Wang HL.Peri-implantitis.J Periodontol. 2018;89(Suppl 1):S267-S290.
7. Sarbacher A, Papalou I, Vagia P, Tenenbaum H, Huck O, Davideau JL. Comparison of Two Risk Assessment Scores in Predicting Peri-Implantitis Occurrence during Implant Maintenance in Patients Treated for Periodontal Diseases: A Long-Term Retrospective Study. J Clin Med. 2022 Mar 20;11(6):1720. doi: 10.3390/jcm11061720. PMID: 35330046; PMCID: PMC8948905.
Vagia P, Papalou I, Burgy A, Tenenbaum H, Huck O, Davideau JL. Association between periodontitis treatment outcomes and peri-implantitis: A long-term retrospective cohort study. Clin Oral Implants Res. 2021 Jun;32(6):721-731. doi: 10.1111/clr.13741. Epub 2021 Mar 27. PMID: 33714224.
Author Response
Dear Reviewer 2
Thank you for the thoughtful and constructive feedback you provided regarding our manuscript, “The Stages and Grades of Periodontitis are Risk Indicators for Peri-implant Diseases”. We hope our revision will be acceptable for your constructive comments.
Keiso Takahashi DDS, PhD, Professor
Division of Periodontics, Department of Conservative Dentistry
Ohu University School of Dentistry
31-1, Misumido, Tomitamachi, Koriyama,
Fukushima 963-8611
Japan

Reviewer 3 Report
In this paper, compare treatment outcomes among patients with perio- 17 dontitis treated with dental implants by experienced periodontists; assess the prevalence of peri- 18 implant diseases: assess factors of implant failures, and their relationships with disease Stage and 19 Grade. It can be improved before publication.
1. The english should be improved.
2. The conclusion cannot reflect the major advantage of this manuscript, I recommend the author should rewrite it.
3. Page 8 Line 279, the "%" should be changed, it make the reader confused.
Author Response
Dear Reviewer 3
Thank you for the thoughtful and constructive feedback you provided regarding our manuscript, “The Stages and Grades of Periodontitis are Risk Indicators for Peri-implant Diseases”. We hope our revision will be acceptable for your constructive comments.
Keiso Takahashi DDS, PhD, Professor
Division of Periodontics, Department of Conservative Dentistry
Ohu University School of Dentistry
31-1, Misumido, Tomitamachi, Koriyama,
Fukushima 963-8611
Japan
In this paper, compare treatment outcomes among patients with periodontitis treated with dental implants by experienced periodontists; assess the prevalence of peri- implant diseases: assess factors of implant failures, and their relationships with disease Stage and Grade. It can be improved before publication.
- The english should be improved.
--> Thank you for your comment. However, we have already used language editing service twice. We will plan to use the service by MDPI if you will inform us more details.
- The conclusion cannot reflect the major advantage of this manuscript, I recommend the author should rewrite it.
--> Thank you for your excellent advice.
We have rewritten according to your comment as below.
- This study demonstrates that severity of periodontitis is a significant risk for development of peri-implant disease at both patient and implant levels. The Stage and Grade classification of periodontitis could be useful tool to predict the treatment outcome of implant therapy of patients with different types of periodontitis. Therefore, further study is required to compare treatment outcome between high- and low-risk groups for peri-implant diseases in a prospective cohort study.
- Page 8 Line 279, the "%" should be changed, it make the reader confused.
--> Thank you for your comment, however, % variance as described in Figure 1 is usually used to indicate the result of principal components analysis.
Round 2
Reviewer 1 Report
Dear authors,
I consider that the article can be published in the present form.
Author Response
Thank you for your understanding and contributions for our manuscript.
Reviewer 2 Report
Revisions to the article entitled “The Stages and Grades of Periodontitis are Risk Indicators for Peri-implant Diseases” By Mikiko Yamazaki 1, Kosaku Yamazaki 2, Yuh Baba3, Hiroshi Ito4, Bruno G Loos5 and Keiso Takahashi 2, *
Briefly, this revised work had aimed at investigating the relationship between peri-implant disease and the severity and complexity of periodontal disease.
The results show different profile (cluster) with high or low risk of peri-implant disease. The strength of this study is the method to determine the clusters. It has the advantage of exploring a field where there are still many risk factors to be confirmed in terms of weight/impact in the development of peri-implant disease.
Based upon our recommendation the bibliography has been up-dated.
General concept remaining comments
The weaknesses still lie in:
The lack of definition of the primary and secondary objectives of the study.
Too many factors (15) are investigated in this retrospective study
Non-calibration of the unique examiner.
Very confusing presentation of results.
Too many off-topic issues in the discussion
General remarks
Since this is the scope of the Journal of Personalized Medicine, I had suggested that it would be more relevant to present only the results at the patient level. You decided not to because “the data from both patient and implant levels are crucial to show the treatment outcome in detail”.
I totally agree with your decision to choose the term implant failure and to present your results according to this term.
Specific comments to authors
The TITLE should indicate that it is a retrospective study.
ABSTRACT: It should be rewritten after suggested revisions. If you choose to not doing revision, the abstract is done.
INTRODUCTION
The introduction has been rewritten with relevant and updated bibliographic references.
The definition of implant success and implant survival should be given with a bibliographic reference. “Implant survival rate or positive outcome was defined as the absence of progressive bone loss as evaluated on radiographs after 6 months, at one year and once yearly afterwards.” Berglundh et al. 2018. This definition was developed and published by a consensus meeting of international experts. You have written the definition of implant failure and success during line 131-138 in material and methods. It would have been more logical in the introduction. As you prefer.
|
Page
|
Line |
Comments |
|
1 |
33-53 |
Please rewrite these sentences and group by topics. |
|
1 |
42 |
Even we all agree with the result of Canullo’s paper, it has no relevance to the purpose of your study since you have eliminated this factor by involving only one experienced practitioner. So you can't analyze this risk factor. Please delete “with surgical skills of dentists being the most significant factor” |
|
2 |
59-62 |
There are too many aims in this study please reduce to one or two aims maximum. For example “the aim of this study was to evaluate the factors of implant failure in periodontal patients and their impact on the prognosis of having a peri-implant disease/failure. «or to evaluate the relationship between implant failure and stage and grade ? |
Material and Methods
The material and method should be restructured and should not include results.
|
Page
|
Line |
Comments |
|
2 |
68 |
Delete “84 patients (325 implants) were enrolled for this study.” These are RESULTS. |
|
2 |
69-72 |
Delete the number of patients between brackets () These are RESULTS to write in result chapter. |
|
2 |
84 |
As you answered, you didn’t realize an intra-individual reliability test of the “trained post-doctoral student” |
|
2-3 |
97-102 |
It is not the issue that the procedure is approved in Japan and supported by scientific data. The bias lies in the wide and justified disparity of procedures. Remove the parameter concerning the BA there is too much disparity in the bone regeneration surgical procedures. It is a huge bias. This parameter should not be analyzed. |
|
3 |
92-107 |
You should put this table 1 in the results chapter In “Diabetes”, specify glycated hemoglobin (HbA1c) less or equal to 7 In smokers, specify if the number of cigarettes smoked is < or > 10 cigarettes/day. Because these are 2 grade modifiers that you used when you classified your patients' periodontitis. |
|
4 |
121 |
Define the method of calibrating the radiographs to reproducibly measure the bone level. |
|
4 |
136 |
Please give the definition of implant survival with reference. |
Results
Please present the patients and implants data in two separated tables.
|
Page
|
Line |
Comments |
|
4 |
156 |
« Mean follow-up period » we misunderstood each other, I would like to know the follow-up time of the patients in your study (mean and standard deviation) in years. |
|
4 |
162-163 |
Please replace these sentence “Implant-supported reconstructions included 50 single crowns, 246 fixed partial dentures, and 29 overdentures.”by “Implant-supported reconstructions included 50 single units and 275 multi-units fixed dentures.” |
|
4 |
165 |
I completely understood the difference of Table 2 and 3. However, you write 2 separate paragraph headings, depending on the level of patient or implant analysis. You simply need to reorganise the presentation of your results.
|
|
4 |
168 |
Add to table 1 that you have to position in Results the Survival rate = 85.7% and success rate = 72.6% |
|
5 |
Table 2 173 |
To facilitate reading and comprehension add to table 2 the total number of patients |
|
5 |
178-179 |
“Mean MBL differed between smokers (2.19 ± 0.82 mm) and non-s…/... test).” Please add in the aims of the study that you will compare MBL between smokers and non-smokers. |
|
5 |
Table 3 184 |
Add to the title of the table “compared to stage III/II/I independently” |
|
5 |
Table 3 184 |
Can you send as appendix these data concerning the other stages and grades ? |
|
6 |
210-214 |
Please clarify and simplify the text. |
Discussion
The discussion had been insufficiently re-structured according to the usual plan which is reminding the reader of the main result of your study and compare it to other data in the literature.
Justify your methodology for example which consisted in an only one experimented periodontist who performed all the surgical procedures in all the 84 patients and point out its limitations, biases and risks of error.
Compare each result with those in the literature.
|
Page
|
Line |
Comments |
|
8 |
303-304 |
“Implant failure may result from single, combined, or multiple factors. It is important to perform patient risk assessment, and to establish a suitable preventive regimen for each patient as personalized medicine. » This should be part of the introduction |
|
8-9-10 |
304 327 348 350 |
There should be no results called (table or figures) in the discussion. Please delete all of them. |
|
8 |
308-309 |
« Chrcanovic et al. reported early failure before occlusal function [14]. Low incidence of early failure in the present study suggests surgical skill in our team was sufficient.” Please delete it is out of scope. |
|
8-9- |
319-331 |
Please delete it is out of scope. |
|
10 |
362 |
« the observation period » please give the number of years or days ? |
|
10 |
376-378 |
“In practice, pocket depth measurements may be unreliable because of the difficulty of probing for implants compared to natural teeth, however, marginal bone loss could be evaluated by dental and panoramic X-ray” This sentence must be placed in the material and methods section. Is it reproducible? |
|
10 |
378 |
« This is the major reason we did not include peri-implant mucositis in this study. » Add to the non-inclusion criteria in the material and methods section |
For all these reasons, my overall recommendation is to Reconsider after Major Revisions.
Author Response
Dear reviewer 2,
Please find enclosed our revised manuscript entitled " The Stages and Grades of Periodontitis are Risk Indicators for Peri-implant Diseases ", which we have rewritten this according to your advice.
We believe that our study makes a significant contribution to the readers for this journal.
Thank you for your consideration. I look forward to hearing from you.
Sincerely,
Keiso Takahashi, DDS, PhD, Professor, guest editor for precision medicine for oral diseases
Division of Periodontics, Department of Conservative Dentistry
Ohu University School of Dentistry
31-1, Misumido, Tomitamachi, Koriyama,
Fukushima 963-8611
Japan

Reviewer 3 Report
Accept.
Author Response
Thank you for your good and useful comments.